# A Combined Analysis of Sociological and Farm Management Factors Affecting Household Livelihood Vulnerability to Climate Change in Rural Burundi

**Risper Nyairo \*** [ID]**, Takashi Machimura \* and Takanori Matsui** [ID]

Graduate School of Engineering, Osaka University, Suita 565-0871, Japan; matsui@see.eng.osaka-u.ac.jp
\* Correspondence: risper@ge.see.eng.osaka-u.ac.jp (R.N.); mach@see.eng.osaka-u.ac.jp (T.M.)

**Abstract:** This paper analyzed the livelihood vulnerability of households in two communes using socio-economic data, where one site is a climate analogue of the other under expected future climate change. The analysis was undertaken in order to understand local variability in the vulnerability of communities and how it can be addressed so as to foster progress towards rural adaptation planning. The study identified sources of household livelihood vulnerability by exploring human and social capitals, thus linking the human subsystem with existing biophysical vulnerability studies. Selected relevant variables were used in Factor Analysis on Mixed Data (FAMD), where the first eight dimensions of FAMD contributed most variability to the data. Clustering was done based on the eight dimensions, yielding five clusters with a mix of households from the two communes. Results showed that Cluster 3 was least vulnerable due to a greater proportion of households having adopted farming practices that enhance food and water availability. Households in the other clusters will need to make appropriate changes to reduce their vulnerability. Findings show that when analyzing rural vulnerability, rather than broadly looking at spatial climatic and farm management differences, social factors should also be investigated, as they can exert significant policy implications.

**Keywords:** questionnaire survey; climate analogues; Factor Analysis on Mixed Data (FAMD); clustering; education; farm management

## 1. Introduction

The Inter-governmental Panel on Climate Change (IPCC) has described vulnerability to climate change as the degree to which a system is susceptible to, and unable to cope with, adverse impacts of climate change [1]. Vulnerability is a combination of the risks people are exposed to and their social, economic, and cultural abilities to cope with the damages incurred [2,3]; therefore, the potential for adaptation is one criterion that may be used to identify key vulnerability of a system [1]. Yet, studies [4,5] have mainly analyzed the biophysical factors that contribute to vulnerability, without accounting for the role played by socio-economic dynamics. Socio-economic conditions have, for instance, been found to profoundly affect food systems through drivers such as soil fertility, irrigation, fertilizer use, demography, and socio-politics [6–8]. Local governments are responsible for providing infrastructure such as water and energy, and hence can play a proactive role in climate change adaptation. The article by [9] elaborates how the development of irrigation farming made possible by federal and provincial governments was a major contributor to community adaptation to climate variability in rural Saskatchewan. Such factors as wealth [10], community organization [11], and access to technology differentiate vulnerability across societies facing similar exposure to climate change [1,7].

On matters scale, while there may be cross-scale interactions due to the interconnectedness of economic and climate systems, local social, cultural, and geographic features may often differ and differentially affect vulnerability levels [12–14]. In fact, social vulnerability has been shown to be a partial product of both social inequalities and place inequalities [15]. IPCC findings [1] clearly illustrate this point by noting that low-latitude less-developed areas are generally at higher risk of climate change impacts and vulnerability due to higher sensitivity and lower adaptive capacity [1]. As [16,17] also noted, attempts to adapt to climate change impacts differed among communities based on geographical location, community attributes, and industrial sectors. IPCC revealed that the impacts of climate variability and extremes are most acutely experienced at the local level. Local level assessment of vulnerability therefore provides a better understanding of where and when to invest and who should make the investment [13,18–21].

The IPCC report further revealed that the African continent is the most vulnerable continent to climate change, given the expected significant reduction in food security and agricultural productivity [1]. Food supply and water resources are some of the sectors in low-latitude areas that are vulnerable to temperature and precipitation changes which result in droughts, decreases in food productivity (especially cereals), and water shortage. There is a general consensus that the sources of vulnerability in Africa are socio-economic in nature and include demography, governance, conflict, and inadequate resources [22]. However, the persistence of drought, which may lead to land cover change, is a potential key impact of climate change [1]. In many parts of the African continent droughts have been experienced more frequently in the last 30 years, and in eastern and southern Africa there is medium confidence that droughts will intensify in some seasons due to reduced precipitation and high evapotranspiration [22]. East Africa, in particular, has experienced temperature increases since the early 1980s. Precipitation in this region is also highly variable in both spatial and temporal terms, but trends show a decrease between the months of March and June [22].

Similarly, Burundi, a country in East Africa, has often experienced the negative impacts of climate variability, especially droughts and flooding. Multi-year droughts have been registered in the periods of 1999–2005, 2007–2008, 2010–2011, and 2016–2017 [23] with dire consequences. With almost 90% of its labor force engaged in agriculture [24], and the agricultural sector making up to 30% of the country's GDP [25], dependence on rain-fed crop production significantly increases the vulnerability of Burundian communities to the negative impacts caused by vagaries of variable weather and climate. Dependence on rain-fed agriculture is a dominant practice among several other countries [26], thus exposing them to drought vulnerability. Apart from the dependence on rain-fed agriculture, the profile of Burundi in terms of economic capacity (agricultural GDP), human resource, and technology is similar to a number of other countries on the African continent. This was illustrated by [27], who ranked Burundi, Somalia, Mali, Chad, Ethiopia, and Niger as countries with high relative vulnerability to drought, based on these similarities.

This paper builds on earlier work by [28], to assess levels of household livelihood vulnerability generated by social processes interacting across geographic scales in two rural communes of Bubanza (Bubanza Province) and Bugabira (Kirundo Province) in Burundi. The objectives are to identify the sociological drivers of household livelihood vulnerability and determine the vulnerability levels among clusters of households in the study area. The aim is to present a human dimension to the analysis of vulnerability of rural livelihoods, which links with existing biophysical vulnerability studies in the two locations.

## 2. Study Background

### 2.1. Climate Analogues Approach

The spatial climate analogues approach is one technique used in aiding climate change adaptation planning by assessing local level social vulnerability of target regions to future climate change. The approach was proposed to overcome the challenges of Global Climate Models, crop models,

and farm system models by presenting field-based realities of the anticipated novel climates using the current spatial variability in climates [29]. Climate analogues are regions whose present climate resembles the predicted climate of another region [30]. In spatial analogues, the analogue region is expected to have climatic conditions similar to those projected for the target region, and is also expected to have similar socio-economic and political conditions [31,32]. In essence, the analogue region is expected to have developed systems adaptable to its climate that the target region can learn from. The spatial climate analogues approach thus resembles contextual vulnerability assessment [32], where it is assumed that nearer locations are more similar than distant locations. Analogue methodologies significantly improve the identification of determinants of vulnerability.

### 2.2. Climate Analogue Analysis in Burundi

In a previous study in Burundi, [28] applied the spatial climate analogue approach in Bubanza Province as the analogue location of Kirundo Province, located approximately 97 km apart. The study showed that farming systems may remain largely similar in the two areas with households keeping similar types of livestock and growing similar kinds of crops despite changed climatic conditions. Slight differences could only be noted in the adoption rates of some improved farming techniques, but these could not be attributed to the difference in climate between the regions. Similar results were reported by [33], who found that factors other than climate were the drivers in farm characteristics. The research by [28] provided useful insights on the expected future of the communes in Burundi but did not account for social variability among households in the two locations, which may serve to exacerbate or ameliorate the predicted negative impacts.

## 3. Materials and Methods

### 3.1. The Study Area

Kirundo province is the northern-most province of Burundi, bordering Rwanda. Geographically, it consists of hills and depressions of Bugesera (88% of total territory), Northeast Bweru (7%), and Buyenzi (5%) [34] regions with moderate to strong slopes. Since the year 1999, the annual evolution has showed a shortening of the rainy season, but with punctually violent rains and an extended dry season. The principal threat on the wet ecosystems is related to over-silting in lower valleys following intense erosion on strong slopes. Kirundo has more than five lakes, including Rweru, Rwihinda, Cyohoha, Kanzigiri, and Gacamirindi, but little access to underground water sources. The lakes Rwihinda and Cyohoha Sud are the nearest to Bugabira commune, with Rwihinda being threatened by excessive evapotranspiration. Bugabira commune is located at approximately 2.3° S and 30.0° E, with elevation ranging between 1000 m and 1500 m above sea level. Rainfall is irregular and bimodal, with average annual precipitation between 800 mm and 1200 mm [25]; the irregularity and reduction in precipitation has already caused drying up of the shallow water sources and reduction in the agricultural production. The mean annual temperature is 20.5 °C. The climate is classified as "moderate" tropical savanna; forest cover is sparse and the area lacks permanent rivers. The ferralitic lithosols present in the area indicate partly-weathered acidic soils of generally low fertility. More fertile organosols [28] are found in the lower valleys. The 2008 census approximated the population of Bugabira Commune at 89,259 persons. Bugabira is documented for over-cultivation and deforestation.

Bubanza Commune, the analogue used for this study, is in Bubanza province and is located at approximately 3.1° S latitude and 29.4° E longitude. The commune is in the Imbo floodplain region near Lake Tanganyika, which has an annual mean precipitation below 1200 mm and a mean annual temperature of 24.0 °C. The commune experiences two main seasons; the wet season runs from September to April, while the dry season runs from May to August. The area experiences long periods of dryness alternating with heavy rainfall and flooding. Bubanza province is distributed in several natural regions because of its backing to the Mumirwa mountain range. Imbo is the major part, followed by Mumirwa and Mugamba. Elevation stretches from 770 m in the Imbo to 2600 m

in the Mugamba region. Soils are lateritic, indicating that they are highly weathered soils with high iron content and low organic matter concentrations. The most common natural vegetation type is savanna. Bubanza is near Nyungwe forest and national park. In terms of hydrology, the Mpanda and Kajeke Rivers and several smaller rivers such as Kidahwe, Nyaburiga, Kadakamwa, and Nyakabingo pass through the commune. The map below shows the two locations (Figure 1). The 2008 census approximated the population of Bubanza Commune at 83,678 persons. Bubanza is known for its paddy fields in the Mitakataka Village.

**Figure 1.** Location of the study areas. Kirundo (Target) and Bubanza (Analogue). Source: Environmental Systems Research Institute; National Oceanic and Atmospheric Administration (ESRI and NOAA).

*3.2. Data Collection*

This study utilized both quantitative and qualitative primary data of 450 households that were collected using a semi-structured questionnaire, as described by [28], for characterizing and clustering the households and also for assessing their vulnerability. A total of 247 households were from Kirundo, the target area, while 203 households were from Bubanza, the analogue. More households were sampled in the target than the analogue location following population estimates of the two locations in order to get representative samples. The questionnaire contained questions on household size, age, education levels, asset ownership, use of farm inputs, food access, land size, whether households identified with community groups and water access and any other changes in resource management and livelihood strategies. This was part of a larger dataset collected for the European Union funded project and implemented by the Consultative Group on International Agricultural Research (CGIAR) program on Climate Change, Agriculture and Food Security (CCAFS). The full list of questions is available as a supplementary table (Appendix A Table A1). The surveys were conducted in April 2012 in the target site and in May 2013 in the analogue location. Households were sampled using the stratified random sampling technique, where they were first sub-divided into two strata according to the administrative unit in which they were located. A number of households were then randomly selected from each unit. Out of the initial more than 1000 questions, including the lower-level dependent ones,

this study extracted the 11 most relevant variables, both quantitative and categorical, as described in the data analysis section.

## 3.3. Data Analysis

The approach for analyzing data was influenced by the objectives and the type of information available. Factor Analysis on Mixed Data (FAMD) [35] and Hierarchical Clustering on Principal Components (HCPC) [36] were combined so as to assess household livelihood vulnerability. FAMD is able to handle both qualitative and quantitative analysis, by applying Principal Component Analysis (PCA) to the quantitative data and Multiple Correspondence Analysis (MCA) to the qualitative data [37]. In this case, MCA was simply a pre-processing step to transform categorical variables into continuous ones. Hierarchical clustering is used in identifying groups of similar observations in a data set, and hence was used to differentiate household profiles.

Out of the total number of original questions obtained by the survey, 107 adaptation-specific variables were first selected (Tables A1 and A2) based on the aspects of demography, infrastructure, household assets, production inputs, and food security [2,38–43]. The variables were then subjected to tests of significance using chi-square [44] and one-way analysis of variance (ANOVA) [45] tests, resulting in only 11 variables (3 quantitative and 8 categorical) for analysis. The 11 selected variables are shown in Table 1. On variable v2 (Lequels), representing rainwater harvesting techniques used, the category "basin" is pools or trenches dig to collect ground water whereas "container" is to store rainwater, and generally "basin" has more capacity. On variable v10 (Monthshung), some farmers suffer food shortage especially in the dry season and the variable represents the severity of food shortage. Variable v11 (Asothe) represents ownership of assets other than crops, animals, radio, cellphone, solar lamp, bicycle, or wheelbarrow.

**Table 1.** Groups and variables selected from the questionnaire survey of households used in Factor Analysis on Mixed Data (FAMD).

| Variable ID | Variable | Description | Type | No. of Levels | Levels (Categorical) Unit (Quantitative) |
|:---:|:---:|:---:|:---:|:---:|:---:|
| v1 | HHeduc | Highest education attained by household head | Categorical | 5 | None/Informal/Primary/Secondary/Tertiary |
| v2 | Lequels | Rainwater harvesting technique used | Categorical | 4 | No/Basin/Container/Others |
| v3 | Ownedland | Amount of land owned | Quantitative | – | Hectares |
| v4 | Ownedfood | Land dedicated to food | Quantitative | – | Hectares |
| v5 | Certseed | Purchase of improved seeds | Categorical | 2 | Yes/no |
| v6 | Buyfert | Purchase of fertilizers | Categorical | 2 | Yes/no |
| v7 | Buypest | Purchase of pesticides | Categorical | 2 | Yes/no |
| v8 | Buyvtmd | Purchase of veterinary medicines | Categorical | 2 | Yes/no |
| v9 | Crdagact | Getting credit | Categorical | 2 | Yes/no |
| v10 | Monthshung | No. of months household is hungry | Quantitative | – | Months |
| v11 | Asothe | Ownership of other assets | Categorical | 2 | Yes/no |

To conduct FAMD, data were first imported into R statistics version 3.4.1, and using the FactoMineR package [46], the computation was run with the number of dimensions retained being eight and without any supplementary variables or individuals. The FAMD result was then used for clustering households in order to characterize them, where a hierarchical clustering by Ward's method was employed. The Ward criterion had to be used because it is based on multidimensional variance as well as Principal Component Analysis. The function HCPC, which is also in the FactoMineR package, was used for clustering, where the appropriate number of clusters was decided using a dendrogram. The HCPC function automatically conducted a chi-square test and output *p*-values of variables that significantly contributed to clustering, with a threshold at 0.05. The ggplot2 [47] and ggpubr packages were used to visualize cluster results.

## 4. Results

### 4.1. Factor Analysis on Mixed Data (FAMD) of Households

FAMD results showed that the first eight dimensions had the cumulative contribution to the variability in the data of 67%, as shown in Table 2. Factor loadings of the dimensions to variables are displayed in Figure 2 in pairs of every two dimensions, where axes show partial contribution to total variability of the data.

**Table 2.** Contributions of the first eight dimensions by FAMD.

| Dimension | Eigenvalue | Percentage Contribution | Cumulative Percentage Contribution |
|---|---|---|---|
| 1 | 2.17 | 13.6 | 13.6 |
| 2 | 1.79 | 11.2 | 24.7 |
| 3 | 1.26 | 7.9 | 32.6 |
| 4 | 1.18 | 7.4 | 40.0 |
| 5 | 1.12 | 7.0 | 47.0 |
| 6 | 1.07 | 6.7 | 53.7 |
| 7 | 1.05 | 6.5 | 60.2 |
| 8 | 1.02 | 6.3 | 66.5 |

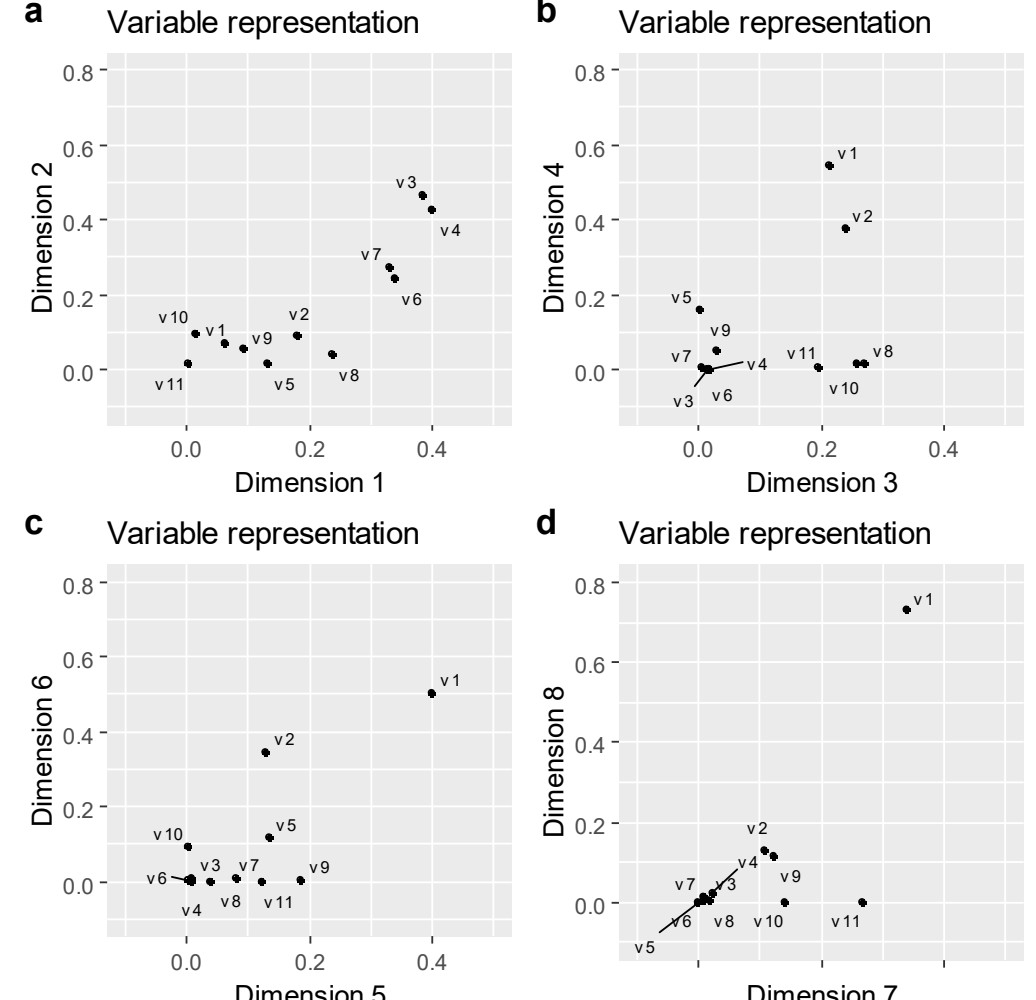

**Figure 2.** Factor loadings of the top eight dimensions on variable groups by Factor Analysis on Mixed Data (FAMD), showing the variable representations by (**a**) dimensions 1 and 2; (**b**) dimensions 3 and 4 (**c**) dimensions 5 and 6; (**d**) dimensions 7 and 8.

In order to characterize the dimensions of FAMD results as indicators of household livelihood vulnerability, significant variables and categories sensitive to the dimensions were summarized in Table 3. In the table, a "+" means the variables were positively related with the dimension, while a "−" means the variables were negatively related with the dimension. Dimension 1 was highly positively related to land used for growing food (v4), the size of land owned (v3), and the purchase of fertilizers (v6) and pesticides (v7). Dimension 2 was positively related to size of land owned (v3) and land used for growing food (v4), but negatively related to purchase of fertilizers (v6) and pesticides (v7). Dimension 3 was positively related to hunger period (v10) and rainwater harvesting (v2) using basin, but negatively related to purchase of veterinary medicines (v8) and ownership of extra assets (v11). Dimensions 6 and 8 were both positively related to tertiary education (v1). However, while dimension 6 was negatively related to container water harvesting (v2) and secondary education (v1), dimension 8 was negatively related to no education (v1). Dimension 7 was positively related to asset ownership (v11) and hunger period (v10), which implies that households on positive coordinates of this dimension were hungry for a longer duration. Dimension 4 was positively related to no education and informal education and negatively related to primary school education and other techniques of rainwater harvesting. Dimension 5 was positively related to secondary education and credit access and negatively related to informal education. Dimension 1 represented farmers' action in improving agricultural production, while dimension 2 represented land ownership. Dimensions 3 and 7 represented affluence levels, while dimensions 6 and 8 represented higher education levels and rainwater harvesting.

**Table 3.** Significant variables and categorical levels significant for the first eight dimensions by Factor Analysis on Mixed Data (FAMD). "+" and "−" denote positive and negative effects on the dimensional scores, respectively.

| Dimension | Significant Variables and Levels |
|:---:|:---|
| 1 | +: Ownedfood, Ownedland, Buyfert, Buypest <br> −: |
| 2 | +: Ownedland, Ownedfood <br> −: Buypest, Buyfert |
| 3 | +: Monthshung, Lequels_Basin <br> −: Buyvtmd, Asothe |
| 4 | +: HHeduc_Informal, HHeduc_None <br> −: Lequels_Other, HHeduc_Primary |
| 5 | +: HHeduc_Secondary, Crdagact <br> −: HHeduc_Informal |
| 6 | +: HHeduc_Tertiary <br> −: Lequels_Container, HHeduc_Secondary |
| 7 | +: Asothe, Monthshung <br> −: |
| 8 | +: HHeduc_Tertiary <br> −: HHeduc_None |

*4.2. Clustering Households Based on FAMD Dimensions*

Clustering based on the first eight dimensions of FAMD yielded five clusters, each with households from both locations, as summarized in Table 4. The number of clusters was decided upon based on a cluster dendrogram. The total number of households in clusters 1 through 5 were 278, 50, 114, 6, and 2, respectively, with clusters 1 and 4 dominated by households from Bugabira (target) and clusters 2 and 3 dominated by households from Bubanza (analogue).

Variables that contributed most to clustering in order of *p*-values were education, purchase of pesticides, purchase of fertilizers, purchase of veterinary medicines, rainwater harvesting, purchase of certified seeds, and access to credit. Among quantitative variables, owned land and land dedicated to food were significant contributors to clustering.

**Table 4.** Number of households classified to clusters per site.

| Cluster | Bugabira (Target) | Bubanza (Analogue) | Total |
|---|---|---|---|
| 1 | 198 | 80 | 278 |
| 2 | 12 | 38 | 50 |
| 3 | 31 | 83 | 114 |
| 4 | 5 | 1 | 6 |
| 5 | 1 | 1 | 2 |
| Total | 247 | 203 | 450 |

Figure 3 shows the distribution of households' score in dimensions with identified clusters, while Table 5 shows the proportions of households per variable within clusters. In order of importance, the significant dimensions for clustering were 1, 2, and 8, followed by 4 and 7. Dimensions 6, 3, and 5 were least important for clustering. Clusters 1 and 3 are almost distinguished in dimensions 1 and 2 (Figure 3a). This result shows that clusters 1 and 3 were defined by dimensions 1 and 2, which represent size of land owned and purchase of fertilizers and pesticides. Extremely high land ownership by only two households that belonged to cluster 5 clearly separated the cluster from other clusters, as shown by panels a and b in Figure 3. Land ownership in cluster 1 was less than the overall average. More than 95% of cluster 1 households did not use pesticides and more than 96% did not use fertilizers, while in cluster 3 more than 84% of households bought pesticides and about 64% used fertilizers (Table 5).

**Table 5.** Significant proportions of households per variable/category within clusters.

| | Variable | Level or Unit | Percent Households or Mean within Clusters | | | | |
|---|---|---|---|---|---|---|---|
| | | | 1 | 2 | 3 | 4 | 5 |
| v1 | HHeduc | None | 0 | 98 | 2 | 0 | 0 |
| | | Informal | 17 | 0 | 18 | 0 | 0 |
| | | Primary | 66 | 0 | 68 | 0 | 0 |
| | | Secondary | 17 | 2 | 12 | 0 | 100 |
| | | Tertiary | 0 | 0 | 0 | 100 | 0 |
| v2 | Lequels | No | 56 | 36 | 18 | 50 | 0 |
| | | Basin | 9 | 14 | 25 | 17 | 50 |
| | | Container | 30 | 48 | 54 | 33 | 50 |
| | | Other | 5 | 2 | 3 | 0 | 0 |
| v3 | Ownedland | Hectares | 1.4 | 1.0 | 1.8 | 4.2 | 41.0 |
| v4 | Ownedfood | Hectares | 1.0 | 0.9 | 1.3 | 1.4 | 23.5 |
| v5 | Certseed | Yes | 29 | 56 | 49 | 83 | 50 |
| | | No | 71 | 44 | 51 | 17 | 50 |
| v6 | Buyfert | Yes | 3 | 22 | 64 | 33 | 50 |
| | | No | 97 | 78 | 36 | 67 | 50 |
| v7 | Buypest | Yes | 4 | 30 | 84 | 17 | 50 |
| | | No | 96 | 70 | 16 | 83 | 50 |
| v8 | Buyvtmd | Yes | 21 | 22 | 68 | 83 | 0 |
| | | No | 79 | 78 | 32 | 17 | 100 |
| v9 | Crdagact | Yes | 19 | 16 | 39 | 17 | 50 |
| | | No | 81 | 84 | 61 | 83 | 50 |
| v10 | Monthshung | Months | 5 | 6 | 6 | 5 | 6 |
| v11 | Asothe | Yes | 19 | 16 | 16 | 33 | 0 |
| | | No | 81 | 84 | 84 | 67 | 100 |

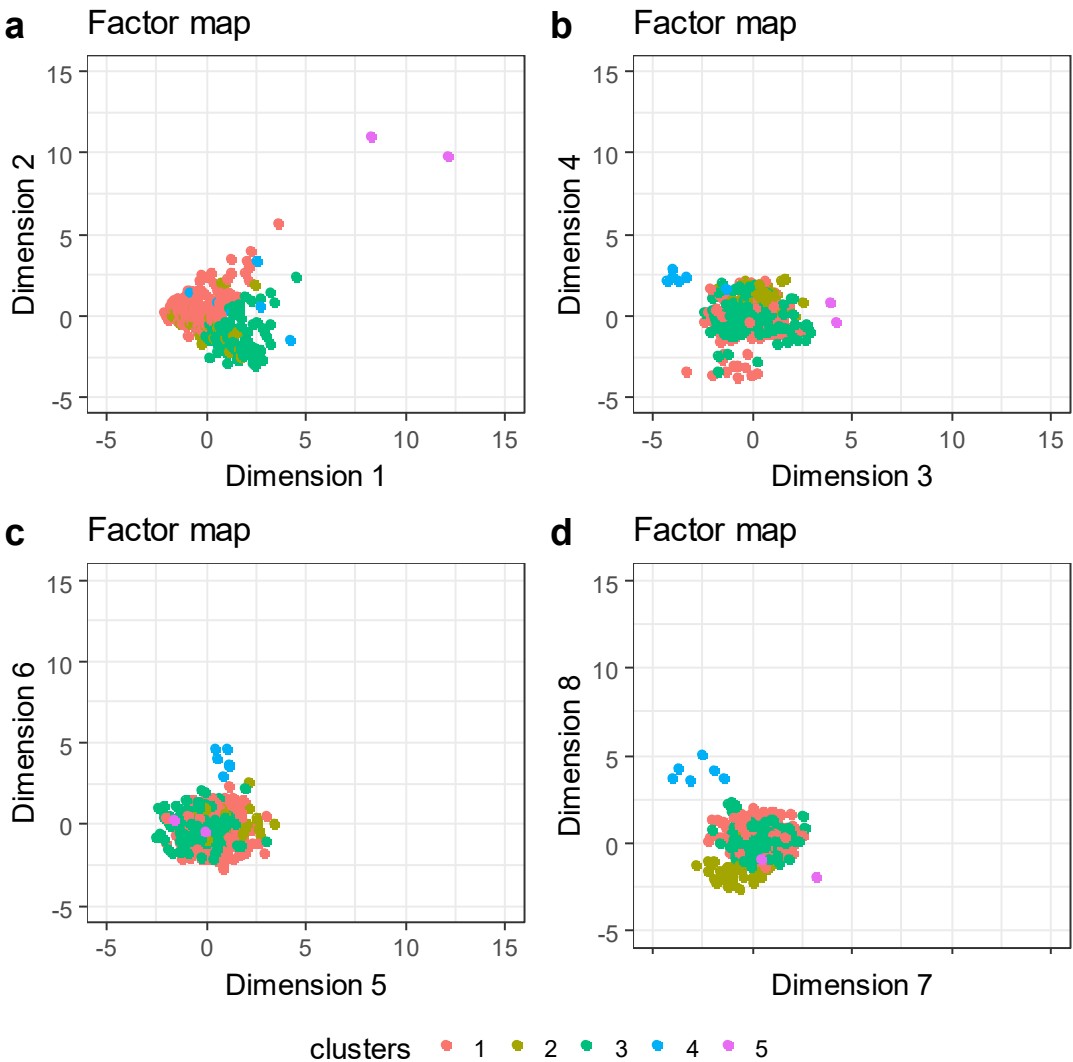

**Figure 3.** Distribution of households' score in FAMD dimensions with identified clusters, showing cluster scores on (**a**) dimensions 1 and 2; (**b**) dimensions 3 and 4; (**c**) dimensions 5 and 6; (**d**) dimensions 7 and 8.

The table below (Table 6) shows dimensional coordinates of cluster centroids. The centroids of cluster 5 had very high positive values on both dimensions 1 and 2, which explains the positioning of cluster 5 households on the factor map (Figure 3). The centroid with the highest negative value was that of cluster 4 on dimension 3.

**Table 6.** Dimensional scores of household cluster centroids.

| Cluster | Dimension | | | | | | | |
|---|---|---|---|---|---|---|---|---|
| | **1** | **2** | **3** | **4** | **5** | **6** | **7** | **8** |
| 1 | −0.68 | 0.48 | 0.02 | −0.22 | −0.01 | −0.06 | 0.10 | 0.17 |
| 2 | −0.09 | −0.51 | 0.42 | 1.30 | 1.00 | 0.25 | −0.91 | −1.61 |
| 3 | 1.42 | −1.17 | −0.12 | −0.17 | −0.44 | −0.17 | 0.27 | 0.10 |
| 4 | 1.71 | 0.85 | −3.5 | 2.27 | 0.84 | 3.93 | −2.80 | 4.07 |
| 5 | 10.2 | 10.4 | 4.01 | 0.25 | −0.85 | −0.07 | 1.83 | −1.40 |

Dimension 8 represented households that either had no education (on the negative coordinates) or those that had attained tertiary education (on the positive coordinates). Panels c and d of Figure 3 show cluster 4 households as having the highest positive coordinates on both dimensions 6 and 8.

This is because all households in this cluster had attained tertiary education. Cluster 2 households had negative coordinates on dimension 8 because almost all (98%) had no education. Cluster 2 households also showed the highest score for dimension 5, which represents getting credit. Cluster 4 was also clearly distinguished on the positive coordinates of dimension 6, which represent tertiary education (panel c of Figure 3). Cluster 4 almost entirely lay on the negative coordinates of dimension 3, which represents either purchase of veterinary medicines or asset ownership. The majority (83%) of the households purchased veterinary medicines (Table 5). Cluster 4 was thus defined by dimensions 8, 6, and 3.

Clusters 1 and 3 largely overlapped in dimension 4 because of the almost similar proportions of households that had attained primary school level education. The wide range of coordinates, both positive and negative, suggests large within-cluster variability in education, with some households attaining secondary education, while others having no education or informal education (Table 5).

The overlap of clusters in dimension 7 but with wide variation in coordinates also suggests large within-cluster variation in the duration of hunger period and ownership of assets, with the exception of cluster 4. Cluster 4 almost entirely lay on the negative coordinates of dimension 7 (panel d of Figure 3), suggesting that households in this cluster either had more assets or they suffered shorter hunger periods.

*4.3. Important Crop and Livestock Species by Sites and by Target Household Clusters*

The questionnaire survey contained two questions on what crop and livestock species were important for households' livelihood. Important crops for livelihood in the target area were wheat, teff, finger millet, peanuts, and sweet potatoes, while in the analogue area the important crops were teff, wheat, banana, barley and sorghum (Figure 4). Important crops in each cluster in the target area are shown in Figure 5, where cluster 5 is excluded because it consisted of only one household. All clusters in the target area except cluster 4 had similar patterns for wheat and teff. Cluster 3 had almost a quarter of the households growing teff and some households growing barley, crops which were more popular in the analogue region.

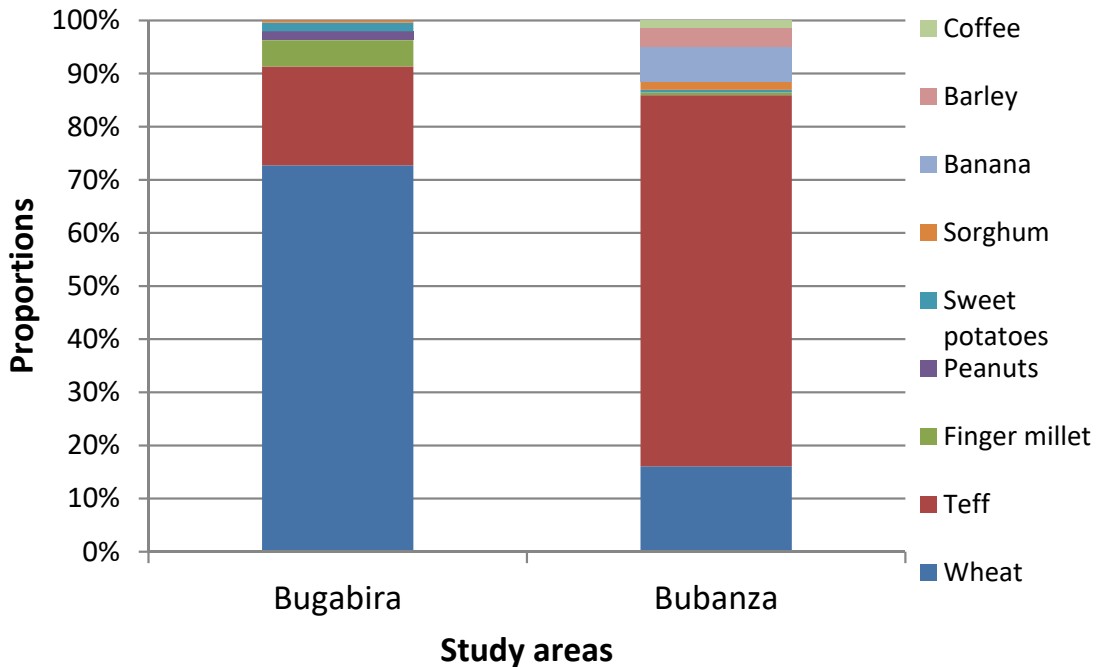

**Figure 4.** Important crops for livelihood in the target and analogue locations.

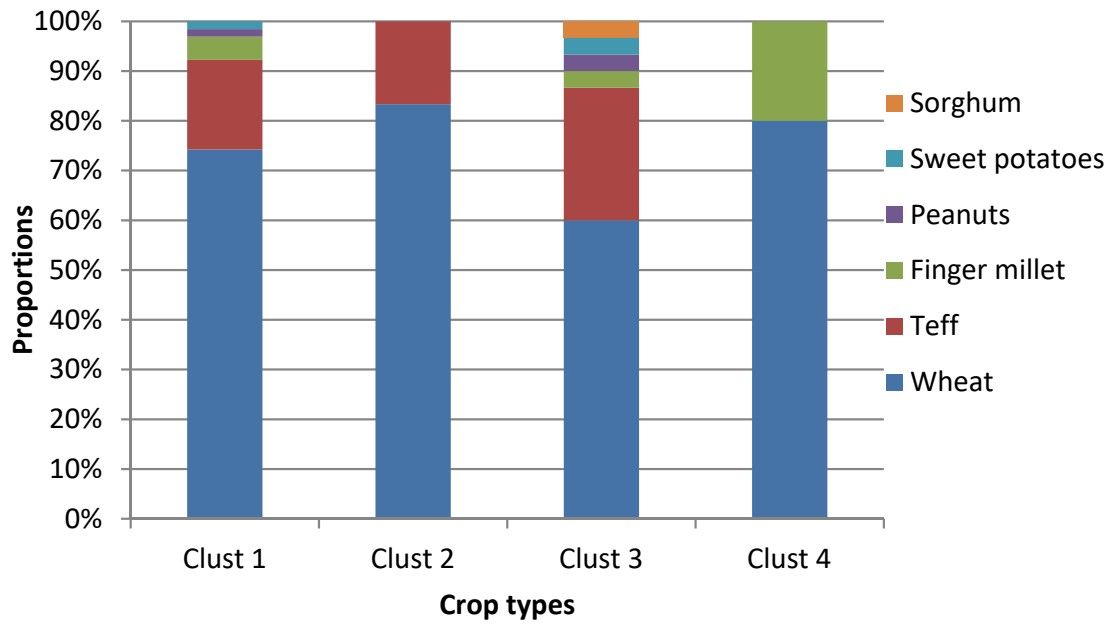

**Figure 5.** Important crops in Bugabira (target site) per cluster.

As for animals, goats, cattle, hens, and horses were important in both sites (Figure 6). Cattle, hens, and horses were almost of equal proportion in the analogue region. The analogue region thus had more hens and horses compared to the target. Households in the target kept more cattle and goats. The important animals per cluster in the target area are shown in Figure 7. Cluster 2 had only goats. Hens and horses were almost equal in cluster 1, while cattle and goats were almost equal in cluster 3. Cluster 4 was dominated by cattle.

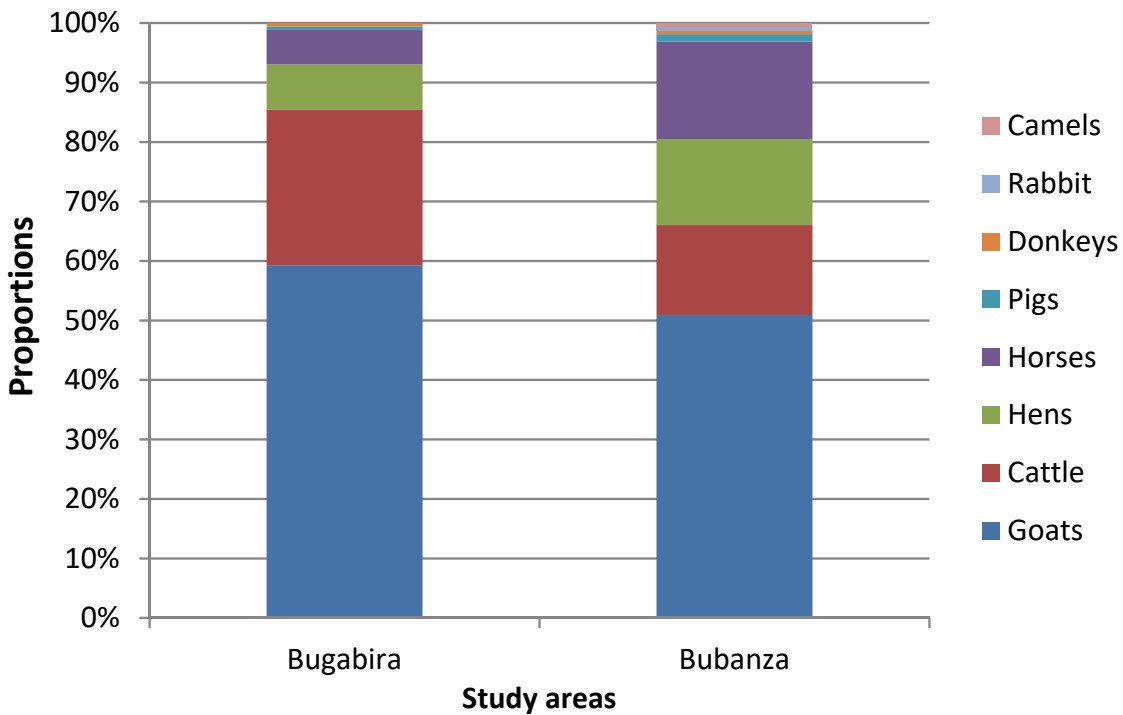

**Figure 6.** Important animals for livelihood in the target and analogue locations.

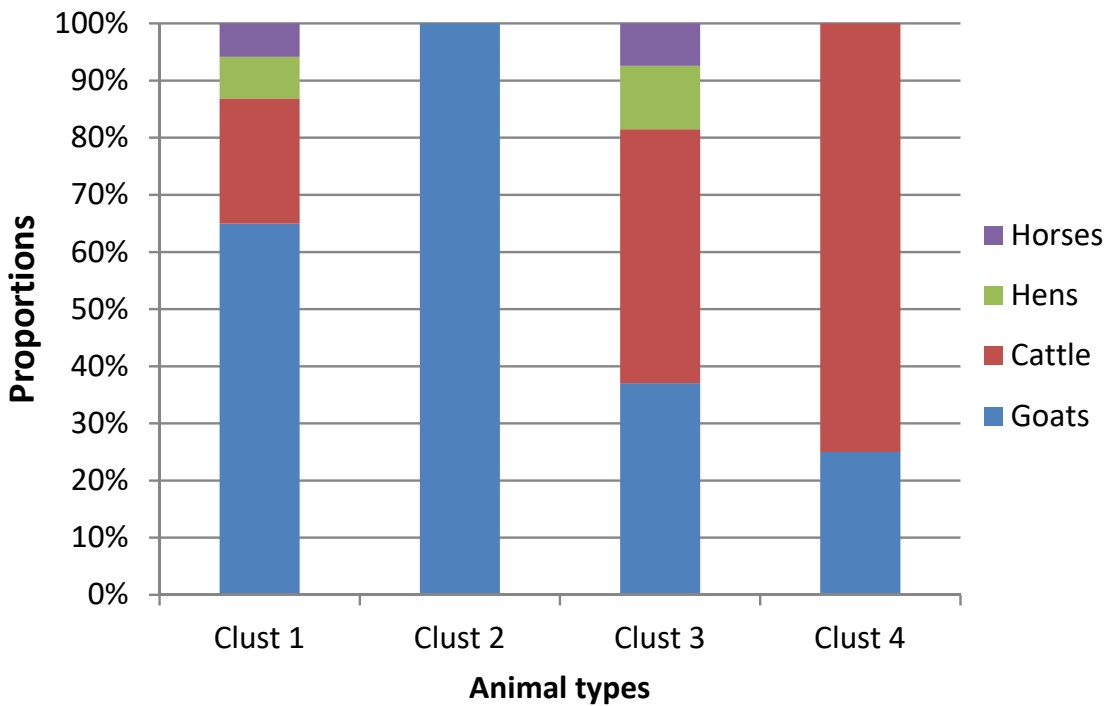

**Figure 7.** Important animals in Bugabira (the target) per cluster.

## 5. Discussion

Human and economic capitals are the representative dimensions of rural sustainability [48,49], while the variables represent the factors that can either constrain or enhance the adaptive capacity of rural households to climatic risks; hence, they form the basis for vulnerability assessment. In the FAMD analysis of this study, the first, second, and third dimensions represented affluence and farmers' actions to improve agricultural production (fertilizer, pesticide, veterinary medicines, and rainwater harvesting), while the fourth to sixth and eighth dimensions mainly represented access to education. Dimension 7 represented households without other assets. The household groups' profiles clustered by the FAMD dimensions reflected these two major aspects and summarized as low- and high-input farming groups (clusters 1 and 3, respectively), and least- and most-educated groups (clusters 2 and 4, respectively). Only two households included in cluster 5 were separated from the others by their extremely large land ownership.

The large variation within clusters in dimensions 3, 4, and 7 suggests that education is closely linked with the duration of hunger periods and ownership of assets. This can be deduced from cluster 4, which had the highest education levels and was negatively related to both dimensions 3 and 7. Education was also the main separator between clusters 2 and 3; given that the majority of households in both clusters were drawn from the same (analogue) location and only had minor differences in the use of certified seeds. It can therefore be concluded that cluster 4 is better placed education-wise to learn new technologies [50,51] and possibly to adapt to the predicted future climate. Already, the cluster grows finger millet (Figure 6), a hardy drought-tolerant crop that remains productive even under low-fertility, low-input systems [52,53], and has access to certified seed and veterinary medicine (Table 5). In addition to learning new technologies, educated populations have the capacity to take advantage of various employment opportunities outside of agriculture, thus reducing their vulnerability [54–56]. In their studies, [57,58] found that respondents acknowledged that getting an education and ultimately securing a job was an adaptive measure to the multiple pressures of climatic and socio-economic changes in the island nation of Tuvalu and in Burundi.

Although cluster 2 had only households with no education, 56% could purchase certified seed, as opposed to only 28% in cluster 1 and 49% in cluster 3, both of which had majority households

attaining primary-level education. Since the majority of households in clusters 2 and 3 were from the analogue location, it would seem that there are "hidden" factors contributing to the purchase of certified seeds in this area. Possible explanations for this are proximity to urban centers, better access to credit (Table 5), and the types of food crops grown in this area. The results of this study revealed that in cluster 3 the number of households accessing credit was twice that of cluster 1. Across eastern and southern Africa, access to credit enables the adoption of new technologies, including the purchase of drought-tolerant maize [59,60]. Maize had already been established as more commonly grown in the analogue than the target region in the study by [28], and it is common practice for maize farmers to use certified seed whenever accessible. Since cluster 1 was credit-constrained, the availability of rainfall becomes critical, hence increasing its vulnerability.

On another note, despite clusters 1 and 3 having almost similar education proportions, cluster 3 had very high proportions of households harvesting rainwater. The large proportion of cluster 3 households who harvested rainwater used containers, meaning the water was for household use. Lack of access to nearby surface water sources as well as general reduction in rain may be driving this practice. Whereas there are several rivers, the province still suffers inadequate access to water, with less than half of the population having access to safe drinking water. The reasons for this are presently unknown, although some reports have linked this to armed conflicts and irregular time-space distribution of rain [24]. This result shows that issues of water access play a role in household livelihood vulnerability, sometimes to the exclusion of education. It also points to the need to evaluate indigenous knowledge and innovation [61] as a source of information when undertaking vulnerability assessment. Indeed, past research [62] reported wider acceptability of indigenous rainwater harvesting techniques by smallholders as opposed to introduced methods.

Cluster 3 also had a high proportion of households using fertilizers, while the use of fertilizers and pesticides among cluster 1 households was almost nonexistent, which may be attributed to the relatively low average land area (Table 5). Similar results were reported by [63]. Ideally, decreasing farm size should trigger intensification, but it is well-known that socio-economic conditions limit the intensification of smallholder farms in Africa. Households with low incomes and limited land are highly risk-averse [64]. Small land size also often means that fallowing cannot be practiced, even when the land is exhausted. Policies that encourage sedentary lifestyles have led to increased vulnerability of communities [65], since now community members are unable to access communal land or migrate in search of fertile land. Migration helps in decreasing the vulnerability of people to climate change [66,67].

The above findings point to the limitation of only using educational level as a determinant of farm management skill. This is in agreement with [49], who cautioned against using formal education to measure the level of skill in farm management, noting that it should be considered together with other constraints in the natural, physical, financial, and social capitals. The study by [65] found that it was a combination of free primary education, provision of roads, and health centers in Botswana that led to a decrease in vulnerability. Cluster 2, for example, had the second highest proportion of households using certified seed second only to cluster 4, despite not having any education. The cluster also had comparatively reasonable access to credit (Table 5). Past studies show that the role of education in aiding credit access is two-pronged; some findings show that higher education increases the probability for credit access, while others show that education decreases demand for credit [68]. In actual sense, the importance of education in accessing credit varies depending on the source of credit. While education may be important for accessing credit from banks and cooperatives, it is not significant when getting money from local community groups, traders, friends, and relatives [69,70].

From the aspect of farming practice, crop and livestock species choice is one of the measures of climate change adaptability of households in the target site. Teff and wheat were found to be the major crops in the two regions, and this pattern was seen in the clusters as well. Wheat generally does very well at 1200 m above sea level, while teff does well across a range of elevations and is both drought and flood-resistant [71]. Sorghum, which was mainly in the analogue region, is an African grass and its C4

photosynthesis pathway allows it to increase net carbon assimilation at high temperatures [72]. It is this drought tolerance that makes sorghum and finger millet important crops in dry regions. Clusters 1 and 3 had diversified in terms of crop types, with cluster 3 already adopting crops grown in the analogue, such as sorghum, suitable in higher temperatures. Crop diversity at the farm level can be achieved either temporally or spatially and helps to reduce risks of fluctuations in climate [67,73]. Cluster 3 in the target site having a similar household profile to majority in the analogue site showed the highest fraction of drought tolerant species selection (Figure 5).

Animals were important for livelihoods in both regions, which is true of African rural households, where livestock is an important capital serving as both a source of food and income [65,74,75]. Livestock is an especially important source of food during dry seasons, because rain-fed crop cultivation is more sensitive to climatic shocks than livestock production [54]. Goats were found to be the majority, and this could be because goats are browsers rather than grazers, and hence require less pastureland, and being heat-tolerant, can survive in a range of environments. In fact, cluster 2 households in the target only had goats, while goats and cattle were of almost equal proportions in cluster 3. It is likely that in future, the proportion of goats will continue to increase across clusters. Clusters 1 and 3 in the target area had diversified animals compared to clusters 2 and 4. The highest dependency by cluster 4 in the target site on cattle, the most beneficial under present climate (Figure 7), suggests the overfitting to present condition, which may weaken the capacity of adaptation to climate change.

## 6. Conclusions

The evaluation of household livelihood vulnerability to climate risks such as drought is important for targeted interventions that improve rural livelihoods and build adaptive capacity. This paper set out to analyze the sources of household vulnerability among smallholder farmers in rural Burundi. The study has shown that members of rural communities are not always vulnerable as a whole based on their location, as it is often assumed, but rather that certain social factors differentiate among levels of vulnerability even within the same location. This was clearly illustrated by cluster 3 households in the target area, who have taken steps to improve their agricultural production and reduce livelihood vulnerability through use of inputs, access to credit, rainwater harvesting, and formal education. Given the findings, cluster 3 households can be said to be the least vulnerable to prevailing and predicted climate conditions, even though more still needs to be done to cover the gap in the use of certified seed and to encourage members to go beyond primary school level.

The findings from this study exert a number of policy implications for reducing livelihood vulnerabilities among smallholders in the study area. Firstly, cluster 1 households will need to adopt practices such as rainwater harvesting and use of improved seed to counter the negative effects of predicted longer and hotter dry seasons. This could be done through designating watering sources and directing run-off to these locations. Water is an important driver of rural economy, as it is used for farming and watering animals. Availability of watering resources reduces distances traveled to access it, thus allowing time for more productive activities. The projected increased rainfall in the study areas can be tapped and used for irrigation to fill the water availability gap. Improving the use of farm inputs such as irrigation and application of integrated fertilizer management can address threats to natural resource-based livelihoods and improve ecosystem service provision. Irrigation, for example, supports crop diversification, since farmers do not have to grow only the crops favored by soil moisture content. Such sustainable intensification should be encouraged through sensitization and awareness creation. Improved seeds that are bred to be drought-resistant and to withstand pest attack should be subsidized and access improved through support to agro-dealers and extension agents. In addition, strengthening the entitlements in terms of land area ownership can promote the use of inputs for improving agricultural production. Smallholders often hold back from investing in land because they feel insecure, especially in many African countries where resource rights are poorly defined and almost never enforced. Interventions such as land consolidation seem to be a solution to

the farm size challenge, although these may be difficult to apply in areas of rapid population growth and lack of alternative off-farm income-generating activities.

Secondly, the most immediate response for cluster 2 should be to offer a mix of education and extension services to the households, as well as cover the gap in the use of improved seed through subsidies. The concept of model farms to demonstrate farming techniques to local farmers, which has been done in countries like Canada and Kenya, can greatly enhance extension service provision and overcome the difficulty of farmers to transition to sustainable practices. In addition, vocational training may be implemented based on locally available resources in conjunction with charity organizations to equip the household members with practical skills that do not require high levels of education. Training can be integrated with the existing credit systems so as to enable sustainable outcomes for the households that access them. Such training should build on existing indigenous knowledge so as to enhance participation and strengthen local capacity.

Finally, reducing household livelihood vulnerability requires multi-faceted, integrated approaches—combining infrastructure, education, and improved land policies to enable adaptive capacity in rural agrarian-based communities. Household characteristics such as affluence and educational status can be seen as buffers that reduce household livelihood vulnerability, but in addition, governments have a part to play. Unequal development may result in certain people benefiting more than others. Lessons can be drawn from the success of Botswana (see [65]) and other African countries. It is the responsibility of the government through the disaster preparedness and management department to put in place contingency plans for eventualities such as drought. Government investment in smallholder crop insurance schemes, for example, has proved important in reducing vulnerability, especially where farms are managed corporately. Doing this will help to achieve one of the targets of one of the United Nations' (UN) Sustainable Development Goals (SDG 2)—end hunger, achieve food security and improved nutrition, and promote sustainable agriculture—in the SDG global indicator framework, which is to double the agricultural productivity and incomes of small-scale farmers by 2030 through secure and equal access to land, inputs, knowledge, financial services, and markets.

**Author Contributions:** Conceptualization, R.N.; T.M. (Takashi Machimura) and T.M. (Takanori Matsui); methodology, R.N. and T.M. (Takashi Machimura); formal analysis, R.N. and T.M. (Takashi Machimura); data curation, R.N.; writing—original draft preparation, R.N.; writing—review and editing, T.M. (Takashi Machimura) and T.M. (Takanori Matsui); supervision, T.M. (Takashi Machimura). All authors have read and agreed to the published version of the manuscript.

**Funding:** This research received no external funding.

**Acknowledgments:** The authors wish to acknowledge Eike Luedeling and the CGIAR Research Program on Climate Change, Agriculture and Food Security (CCAFS) for support in data acquisition and Institut des Sciences Agronomiques du Burundi (ISABU) for logistic support. Special thanks go to Concern Worldwide, Burundi and all households who participated in the survey. The authors also thank all 3 anonymous reviewers whose comments led to the improvement of the manuscript.

**Conflicts of Interest:** The authors declare no conflict of interest.

## Appendix A

**Table A1.** Full questionnaire design.

| Questionnaire Section | Description | Number of Questions | Number of Variables |
|---|---|---|---|
| 1 | Household respondent and type | 6 | 6 |
| 2 | Demography | 4 | 4 |
| 3 | Sources of livelihood security | 3 | 128 |
| 4 | Crop, farm animals, and tree management changes | 7 | 607 |
| 5 | Food security | 2 | 24 |
| 6 | Land and water | 11 | 42 |
| 7 | Input and credit | 11 | 32 |
| 8 | Climate and weather information | 1 | 50 |
| 9 | Community groups | 5 | 44 |
| 10 | Assets | 1 | 56 |
| 11 | Constraints to production | 2 | 18 |

**Table A2.** Selected questions for analysis.

| Section | Variable Group | Variable | Levels or Unit |
|---|---|---|---|
| 1 | Age | Age of household head; Age of respondent | Age in years |
| 1 | Sex/Education | Sex of the household head; | Male/Female |
| | | Highest education attained by household head | No education/Informal/ Primary/Secondary/Tertiary |
| 2 | Household size | Total household size<br>Total household males<br>Total household females<br>Males below 5 years<br>Females below 5 years<br>Males between 5 and 15 years<br>Females between 5 and 15 years | Number |
| 6 | Rainwater harvesting | Practice rainwater harvesting | Yes/No |
| | | Use of harvested rainwater | Irrigation/Household/ Livestock |
| | | Practice irrigation | Yes/No |
| | | Type of irrigation practiced | Basin/Container/Dam |
| 10 | Assets | Do you own a radio? Cellphone? Bicycle? Motorcycle? Solar panel? Machete? Wheelbarrow? Spade? Watering cans? Oil lamp? Large battery? Fishing net? Bank account? Cattle? Goats? Sheep? Poultry? Other assets? Water storage tank?<br>Improved house, e.g., brick/concrete<br>Improved roofing<br>Separate animal structure | Yes/No |
| 6 | Land access | Size of land owned<br>Land rented<br>Communal land<br>Total land access<br>Land owned dedicated to food<br>Land rented dedicated to food<br>Communal land dedicated to food<br>Land owned for industrial purpose<br>Land owned dedicated to grazing<br>Land owned dedicated to trees<br>Land rented dedicated to trees<br>Land owned for aquaculture<br>Degraded land<br>Tanks for water harvesting | Hectares |
| 6 | Water sources | Dams or water ponds<br>Water inlet | Yes/No |
| 7 | Inputs | Purchase of certified seed<br>Purchase of fertilizers<br>Purchase of pesticides<br>Purchase of veterinary medicine<br>Access to agricultural credit | Yes/No |
| 9 | Group membership | Tree planting, Fish pond, Fishing, Forest product collection, Water catchment management, Soil improvement activities, Crop substitution, Irrigation, Savings and credit, Marketing products, Productivity enhancement, Seed production, Vegetable production<br>Any other group | Yes/No |
| 9 | Support groups | Help from friends<br>From government<br>From politicians, MPs<br>From NGOs<br>From religious organizations/church<br>From local community group | Yes/No |
| 9 | Association membership | Purpose of association you are a member of Association benefits | Agriculture/Marketing/Savings and credit/ Other support Access to credit/ Shared manpower/ Common purchase of inputs/ Easier access to information/ Coordinated agricultural sales |
| 9 | Association size | Total association membership<br>Number of men<br>Number of women | Number |
| 5 | Food source/scarcity | Source of food in January, February, March, April, May, June, July, August, September, October, November, December | Mainly from own farm/ Mainly from off-farm (purchase, aid) |
| | | Shortage of food in January, February, March, April, May, June, July, August, September, October, November, December | Yes/No |

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
