# Peer review of "A Combined Analysis of Sociological and Farm Management Factors Affecting Household Livelihood Vulnerability to Climate Change in Rural Burundi"

_sustainability, doi:10.3390/su12104296_

Round 1
Reviewer 1 Report
I congratulate Risper Nyairo, Takashi Machimura, Takanori Matsui for a considered study, well thought through and executed. It had a strong sampling, comparison, and presentation of results. Clear descriptions of study sites were made and a broad number of references cited.
I could not verify the assertion that studies (line 36) cited in endnotes 'have been narrow' and too dependent on the analysis of biophysical factors only in determining vulnerability to climate change.
The overall conclusion that assessments of local-level vulnerability provide a better understanding of investment decisions, and more focused targeting for adaptative capacity.
I thought the correlations between education and other variable dimensions of the study provoking.
I hope the study's findings lead to clearer policy interventions and focus.
Reviewer 2 Report
Dear Authors,
thank you for giving me the opportunity to review this interesting article. I think the article is promising, however, I have some suggestions that I hope will help you in further improving the paper.
Abstract: the abstract focus too much on the analysis and too less on the context, please provide an abstract which covers really shortly all the aspects: broader topic, why it is important, specific focus on the gap you fill, method, results, and contributions.
Introduction: When you introduce the context of your analysis (Burundi) you should focus more on the characteristics that make it similar to other contexts in order to increase the generalizability of your study. I would also shorten the introduction by moving lines from 87 to 96 in a theoretical background section that you still don't have. It does not need to be too long the theoretical background section but at least it can help to frame better your research. I would explain better, in the introduction, the exploratory nature of your study to justify why you don't have a specific theory behind your study but just a stream of literature. I would also stress better the gap you are going to fill in your study. This can also help you in developing the discussion and the contributions.
Method: the method is appropriate with your aim.
Discussion and conclusions: I think you should enrich the discussion with further comparisons with previous academic literature. In the conclusion, the theoretical contributions can be improved.
Writing: please check for typing mistakes, both in terms of grammar mistakes and typing mistakes. Line 186, for instance, you have brackets without anything inside.
Finally, I suggest you some papers which may help you to develop your theoretical section and discussions. The first one is from this journal and would stress the link with the editorial line of this journal while the other two ones are more strictly related to your topic.
- De Bernardi, P., Bertello, A., & Venuti, F. (2019). Online and on-site interactions within alternative food networks: Sustainability impact of knowledge-sharing practices. Sustainability, 11(5), 1457.
- Morton, J. F. (2007). The impact of climate change on smallholder and subsistence agriculture. Proceedings of the national academy of sciences, 104(50), 19680-19685.
- Altieri, M. A., & Koohafkan, P. (2008). Enduring farms: climate change, smallholders and traditional farming communities (Vol. 6). Penang: Third World Network (TWN).
Reviewer 3 Report
The following paper analyse socio-economic as well as farm management practices in a target and an analogue location, profiling the “good practices” to be undertaken in order to reduce climate changes vulnerability in rural Burundi. Highlighting that these don’t depend only on the geographical location and farm management, as it’s commonly thought, but are strongly related also on socio-economic factors. The work is generally well done and interesting, with the need of only some improvement to clarify it.
Introduction:
LINES 103-107: It seems more appropriate for an abstract than an introduction
Materials and methods:
LINES 111-112: Since percentages are used, I suggest citing all the geographical features of the territory. Here you explain the 95% of its composition, what is the remaining 5% occupied for?
LINE 118: In figure 1, Kirundo province and Bubanza commune are shown. I suggest showing Bugabina commune instead of Kirundo province, since later in the article you always refer to commune and not province.
LINES 138-140: Hydrology situation isn’t analogue to Kirundo one, this difference could affect the result?
LINES 151-152: Why a different distribution of characterization with more households in target site?
LINES 158-159: Since data were collected in different seasons, are responses of the analogue site reliable? May is dry season and this factor could affect the results in the analogue commune. If you do that because the target commune is generally dryer, due also to its hydrological condition, then it is better to explicit this reason.
Results:
Chapter 3.1: no consideration on Dimension 4 and 5.
LINE 212: In materials and methods you don’t cite correlation test, or in this case “correlated” is used as synonym of “related”?
LINES 213-214: From table 1 I see that food owned (v4) has a positive relation with Dimension 1, too.
LINE 215: As above.
LINE 238: From what test was obtained p-value?
LINE 289: Why don’t you show results for the analogue site, too?
LINE 299: As above.
Discussion:
LINE 306: There are no considerations for the remaining dimensions?
LINE 341: “lack of nearby surface water”…but in lines 328-329 you said that this cluster is overall composed by households in analogue location, which you wrote in line 139 small rivers pass through. Are these rivers dry in the study season? If yeas, it would be clearer to say it.
LINE 382: “;” instead of “.”
LINE 387: No considerations about analogue? Since cluster 3 is like the analogue (LINE 378), this means that livestock composition isn’t dependent on the household profile depicted above? Or, on the contrary, the analogue has a heterogeneous livestock composition too, and so this characteristic could be considered the “future”, against what you predict in LINE 387?
I think that discussion is well done, however I would appreciate a more detailed comparison with the analogue location, which should be better described, briefly highlighting why this location could be consider adapted to its own climatic conditions. Because live in a determinate condition doesn’t mean necessarily being well adapted to it. Summarizing, I suggest to explicit the systems (cited in line 83) that turn Bubanza in a good analogue.
Conclusions are well done, clear and I think very interesting.
Round 2
Reviewer 2 Report
The authors have implemented the parts suggested by the reviewers. I would suggest a final proofreading.
Author Response
Point 1: The authors have implemented the parts suggested by the reviewers. I would suggest a final proofreading.
Response: The manuscript has been proofread by authors. Minor revisions were made to the introduction and conclusion sections. Some figures were formatted. Grammatical errors were corrected.